# Chemical Constituents from *Ficus sagittifolia* Stem Bark and Their Antimicrobial Activities

**DOI:** 10.3390/plants12152801

**Published:** 2023-07-28

**Authors:** Olayombo M. Taiwo, Olaoluwa O. Olaoluwa, Olapeju O. Aiyelaagbe, Thomas J. Schmidt

**Affiliations:** 1Department of Chemistry, University of Ibadan, Ibadan 200284, Nigeria; omosalewa.olaoluwa@gmail.com (O.O.O.); oaiyelaagbe@gmail.com (O.O.A.); 2University of Münster, Institute of Pharmaceutical Biology and Phytochemistry (IPBP), PharmaCampus, Corrensstrasse 48, D-48149 Münster, Germany

**Keywords:** *Ficus sagittifolia*, flavonoids, dihydrobenzofuran, coumarins, apocarotenoid, antimicrobial activity

## Abstract

The phytochemical investigation of the ethylacetate fraction of an ethanolic extract obtained from the stem bark of *Ficus sagittifolia* (Moraceae) led to the isolation of four flavonoids: (2*R*)-eriodictyol (**1**), 2′- hydroxygenistein (**2**), erycibenin A (**3**), and genistein (**4**); a dihydrobenzofuran: moracin P (**5**); a coumarin: peucedanol (**6**); and an apocarotenoid terpenoid: dihydrophaseic acid (**7**). These were identified via 1D and 2D nuclear magnetic resonance spectroscopy (NMR) and ultra-high-resolution liquid chromatography–quadrupole time-of-flight mass spectroscopy (UHPLC-QTOF MS). Moracin P (**5**) is being reported for the first time in the genus *Ficus,* while the others are known compounds (**1–4** and **6–7**) isolated previously from the genus but being reported for the first time from the species *F. sagittifolia*. Their antimicrobial activity against various pathogens (five bacteria: *Escherichia coli*, *Klebsiella pneumoniae*, *Pseudomonas aeruginosa*, *Staphylococcus aureus*, and *Salmonella typhi*; two fungi: *Aspergillus niger* and *Candida albicans*) was tested. The mixture of genistein and moracin P (**4**+**5**) exhibited strong activity against *K. pneumoniae* (MIC < 0.0039 mg/mL), whereas dihydrophaseic acid (**7**) was the most active against *P. aeruginosa* and *A. niger* (MIC = 0.0078 and <0.0039 mg/mL, respectively). These compounds might be considered potential antimicrobial agents with the potential to be starting points for the development of antimicrobial drugs.

## 1. Introduction

Despite the many existing antibiotics, infectious diseases are consistently ranked second among causes of death worldwide [1]. Antimicrobial resistance is an increasingly serious problem in the twenty-first century, with reports stating that more than 10 million deaths annually could arise by 2050 [2]. The pursuit of new compounds with therapeutic potential for infectious diseases is paramount, and medicinal plants are a promising reservoir of drug or lead compounds [3].

Medicinal plants contain bioactive organic chemical compounds, which play a defensive role for the plants but may also be utilized in medicine, e.g., against major chronic diseases in host–metabolic and genetic dysfunctional disorders as well as infectious disease [1,4]. Medicinal plants are rich sources of a wide variety of secondary metabolites, such as phenolic compounds, flavonoids, alkaloids, tannins, and terpenoids, many of which have been found to have antimicrobial properties [5,6]. From one major group, phytochemicals that have been studied extensively for antimicrobial activities are the flavonoids [7,8]. Flavonoids are also known for their antioxidant, anti-inflammatory, anti-cancer, antiallergic, antimicrobial, and hepatoprotective properties [9]. To date, many flavonoids have been reported to possess antibacterial activities against plant pathogens, which could be applied to fight human pathogens as well [10].

*Ficus sagittifolia* Warburg. ex Mildbraed & Burret is a plant from the Mulberry family (Moraceae). It is an epiphyte shrub/tree, often on oil palms about 30 feet high. It is native to Sub-Saharan West Africa [11]. Traditionally, bark and leaves are used to treat pulmonary disorders and stomach disorders, respectively. The bark is edible and also taken against colic [12]. Species in the genus *Ficus* (figs) are a rich source of phytochemicals, such as flavonoids, triterpenoids, phenolics, coumarins, anthocyanins, and glycosides. These are responsible for their various biological activities, including antimicrobial, antioxidant, antiplasmodial, antidiarrheal, antidiabetics, anthelmintic, anti-inflammatory, anti-ulcer, antipyretic, and gastroprotective activities [13]. The leaves of *F. sagittifolia* have been investigated for their phytochemical constituents, which led to the isolation of stigmasterol glucoside, prunetin, derrone, and alpinumisoflavone. Stigmasterol glucoside had good activity against *Pseudomonas aeruginosa,* while prunetin and alpinumisoflavone were both very active against *Helicobacter pylori*. Prunetin also showed good inhibition against *Aspergillus niger* among others [14]. Our interest in *F. sagittifolia* as a potential source of antimicrobials led us to investigate the phytochemical constituents present in *F. sagittifolia* stem bark and their antimicrobial activity.

## 2. Results and Discussion

### 2.1. Chemical Constituents from F. sagittifolia Stem Bark

Chromatographic separation of the ethylacetate fraction of an ethanol extract obtained from *F. sagittifolia* stem bark afforded seven known compounds (**1**–**7**; Figure 1). They were identified via detailed spectroscopic analysis and comparison of their spectroscopic data with those reported in the literature as (2*R*)-eriodictyol (**1**) [15,16,17], 2′-hydroxygenistein (**2**) [18,19], erycibenin A (**3**) [20,21], genistein (**4**) [22,23], moracin P (**5**) [24], peucedanol (**6**) [25,26,27], and dihydrophaseic acid (**7**) [28,29]. Compounds **1** and **2** as well as **4** and **5** were obtained as mixtures. The approximate composition was 30:70 for **1** and **2** and 80:20 for **4** and **5**, as estimated from their ^1^H-NMR spectra.

It should be noted that the *R*-enantiomer of Eriodictyol (**1**) was unambiguously identified via the circular dichroism (CD-) spectrum obtained with the mixture of **1** with the non-optically active **2**. The positive Cotton effect (CE) at 261 nm is in agreement with literature data for other 2*R*-configured flavanones [30,31,32]. This was unambiguously confirmed through computational simulation of the CD spectrum (see Figure 2) using time-dependent density functional theory (TDDFT) calculation. The simulated spectrum for the (2*S*)-enantiomer is opposite in sign to the experimental spectrum, which, in turn, matches very well with the theoretical spectrum of the (2*R*) form.

In the case of the apocarotenoid dihydrophaseic acid (**7**), the NMR data were in good agreement with those published [28] so that there could be no doubt on the assignment. However, the ^1^H-NMR data reported in the literature require small revisions (compare Appendix A, Appendix A): the resonances of the protons at positions 7 and 8 have to be exchanged, i.e., H-7 resonates at a lower field (*δ* 7.99 ppm) due to its more unshielded position in the *α,β,γ,δ*-unsaturated carboxylic acid side chain than H-8 (*δ* 6.53 ppm). Furthermore, the multiplicities and coupling constants of H-2*α*, 2*β*, 4*α,* and 4*β* as well as H-3 require revision. The latter is a *dddd* appearing as a pseudo-“triplet of triplets” with 10 and 7 Hz, whose coupling constants with H-4*α* and H-2*α* as well as with 2β and 4*β* (both in the *β*-position) are almost identical to 7.0 in the case of the former and 10.2 as well as 10.9 Hz in the case of the latter pair. In addition to these regular *^3^*J couplings, H-2*α* and H-4*α* share a ^4^J “W”-type coupling of 1.8 Hz. Another “W”-coupling of 2.2 Hz is observed between H-4*β* and H-12α (=12*β*). All those coupling constants are in full agreement with the dihedral angles and relative orientations of these protons in 3D molecular models of compound **7** in the depicted configuration. In addition to the mentioned couplings, an allylic *^4^*J coupling is observed between the methyl group at position 15 and the proton H-10, which, therefore, appear as a doublet and a quartet (*J* = 1.3 Hz), respectively.

The CD spectrum of **7** (see Figure 3) was in agreement with the reported data of [29], with a prominent negative CE at 265 nm (Δε = −3.8; reported: 266, −2.8 [29]). However, the configuration published in this work [29] was different from that reported later [28,33]. It should be noted that another work [33] also reported a negative CE of (−12 at 276 nm), but the depicted spectrum was of poor quality. Nevertheless, it resembles that recorded in the present study and the data reported in [29]. However, there appears to be some uncertainty in the structural/spectral assignment so that, in order to obtain an unambiguous assignment, we also simulated the CD spectrum of **7** in the depicted configuration (corresponding to [28,33]) via TDDFT in a similar manner as above. The resulting simulated spectrum obtained with the lowest-energy conformer is in agreement with the reported data in that it shows a negative sign but with two CEs around 300 and 250 nm (Figure 3). Although, the fit with the experimental spectrum is not as good as in the case of compound **1** above, the simulation matches the depicted absolute configuration much better than the enantiomer would so that the assignment of absolute configuration is confirmed.

The genus *Ficus* is characterized by flavonoids, coumarins, terpenoids (triterpenoids and sesquiterpenoids), and alkaloids [34]. In the present study, seven compounds (**1**–**7**) were isolated and identified from the stem bark of *F. sagittifolia*, which were classified as flavonoids (**1**–**4**), a dihydrobenzofuran (**5**), a coumarin (**6**), and an apocarotenoid terpenoid (**7**). Species in the genus *Ficus* are rich sources of flavonoids, with isoflavones as the major reported class [23,35,36]. This is also reflected in the present study by compounds **1**–**4**, i.e., the majority of the isolated compounds from *F. sagittifolia* stem bark were flavonoids, most of which (**2**–**4**) belong to the isoflavonoid subclass. All the isolated compounds were obtained and identified from this species for the first time. Quite noteworthily, (2*R*)-(−)-Eriodictyol (**1**) is much less frequently found in plants than the (2*S*)-(+)-enantiomer (41 entries vs. >3000 entries in the Chemical Abstracts) [37]. Thus, the *S*-form has previously been reported from several *Ficus* species and other Moraceae [17,38,39,40,41], whereas the *R*-form has now been identified for the first time with certainty in the genus *Ficus* and the family.

Furthermore, in this study, the dihydrobenzofuran derivative moracin P (**5**) was isolated for the first time from the genus *Ficus* (Moraceae). It is not known whether it exists in other *Ficus* species but has previously been isolated from the herbal drug *Mori Cortex Radicis* (obtained from *Morus alba*, Moraceae) [24]. In view of the many chemical studies on other Ficus species in which the compound was not found, compound **5** might be considered a characteristic component and, possibly, a chemotaxonomic marker for *F. sagittifolia*. The coumarin derivative **6** is reported for the second time in the genus *Ficus*, the first report being from *F. glumosa* [26], which supports the taxonomic position of *F. sagittifolia* in the genus with a close phylogenetic relationship to *F. glumosa*. Compound **7** is a derivative of the plant hormone abscisic acid that can be classified as apocarotenoid terpenoid, which has previously been reported from *F. consociata* [33] and *F. hirta* [42]. The discovery of compound **7** in *F. sagittifolia* might also point towards closer relationships with these species. Overall, these findings add to the knowledge on the chemical constituents of *F. sagittifolia* and provide additional information for chemosystematic research on the genus *Ficus*.

### 2.2. Antimicrobial Activity of Isolated Compounds from F. sagittifolia Stem Bark

Currently, there is growing interest in plants as sources of antimicrobial agents due to their use from ancient times. Even today, a significant part of the world’s population, particularly in developing countries, depends on plants for the treatment of infectious and non-infectious diseases. Determination of the minimum inhibitory concentration (MIC) as well as minimum bactericidal or fungicidal concentrations (MBCs, MFCs) is relatively simple first measure to estimate the potential effectiveness of antimicrobial agents and may be predictive of therapeutic outcome.

The mixture of (2*R*)-eriodictyol (**1**) and 2′-hydroxygenistein (**2**) showed inhibition against the test organisms, with MICs ranging from 0.25 to 0.0625 mg/mL and MBC/MFC ranging from 0.25 to 0.125 mg/mL. The mixture of **1** and **2** was active against the growth of *E. coli* and *A. niger* at a low MIC value of 0.0625 mg/mL. (2*S*)-Eriodictyol has been reported to possess antimicrobial, antioxidant, anti-inflammatory, anticancer, neuroprotective, cardioprotective, antidiabetic, antiobesity, and hepatoprotective activities [43,44,45]. Erycibenin A (**3**) had MIC values of 0.25–0.0625 mg/mL and MBC/MFC values ranging from 0.25 to 0.125 mg/mL. For erycibenin A, an MIC of only 0.0313 mg/mL was required to inhibit the growth of *E. coli* and *A. niger*. The mixture of genistein (**4**) and moracin P (**5**) inhibited the growth of the test organisms with MICs ranging from 0.25 to 0.0039 mg/mL and MBC/MFC values ranging from 0.25 to 0.0313 mg/mL. This binary mixture of **4** and **5** displayed relatively good activity against *K. pneumoniae,* with an MIC and MBC of 0.0039 and 0.0313 mg/mL, respectively. Genistein was also reported to possess promising antimicrobial activity against the growth of *Staphylococcus aureus*, *Streptococcus pasteurianus*, *Bacillus cereus*, and *Helicobacter pylori,* as reported by [46]. Peucedanol (**6**) inhibited the growth of the test organisms with MIC values ranging from 0.25 to 0.0313 mg/mL and MBC/MFC values ranging from 0.25 to 0.0039 mg/mL. Peucedanol inhibited the growth of *C. albicans* and *A. niger* at an MIC of 0.0625 and 0.0313 mg/mL, respectively. Dihydrophaseic acid (**7**) inhibited the growth of the test organisms, with MICs ranging from 0.25 to 0.0039 mg/mL and MBCs/MFCs ranging from 0.25 to 0.125 mg/mL. Dihydrophaseic acid (**7**) displayed its highest activity against the growth of *P. aeruginosa* and *A. niger* with MICs of 0.0078 and 0.0039 mg/mL and MBC/MFC of 0.25 mg/mL, respectively. The reference antibiotic gentamycin and ketoconazole inhibited the growth of the test organisms with MIC and MBC values ranging from 0.01 to 0.005 mg/mL.

Overall, compounds **1–7** showed moderate activity against the test organisms, as compared to the positive controls used, but in some specific cases, the activity was comparable to that of the established antibiotics (Table 1). These particular compounds might, hence, represent interesting starting points for further investigations/development.

## 3. Materials and Methods

### 3.1. General Instrumental Methods

The vacuum liquid chromatography was performed using a TLC-grade binder-free silica gel (silica gel H) on a sintered glass funnel with fritted disk grade G3.

Preparative HPLC separations were performed on a JASCO (Groß-Umstadt, Germany) preparative HPLC system using a Reprosil 100 C-18 preparative reversed-phase column (5 μm, 250 mm × 20 mm, Macherey-Nagel, Düren, Germany) with binary gradients of the mobile phase consisting of water and methanol or acetonitrile containing 0.1% trifluoroacetic acid. ChromNav, Version 1.18.06, software was used to record chromatograms at 200, 254, 300, and 365 nm.

The NMR spectra, 1D (^1^H and ^13^C) and 2D (^1^H/^1^H-COSY, ^1^H/^13^C HSQC and HMBC), were recorded on a 600 MHz Agilent DD2 NMR spectrometer (Agilent Technologies, Santa Clara, CA, USA). All samples were dissolved in deuterated methanol (methanol-d4) and measurements performed at 298 K. Spectra were referenced to the signal of the solvent’s methyl group at δ = 3.310 and 49.000 ppm, for ^1^H and ^13^C, respectively.

The CD spectra were recorded in methanol at room temperature and at concentration of 0.2 mg/mL with a Jasco (Groß-Umstadt, Germany) J-815 CD-spectropolarimeter using a 0.1 cm quartz cuvette.

UHPLC-QTOF MS analysis (further on termed LC-MS) was performed on a Dionex Ultimate 3000 RS liquid chromatography system (Idstein, Germany) with a Dionex Acclaim RSLC 120 C18 column (2.1 × 100 mm, 2.2 μm) using a binary gradient of water and acetonitrile, both with 0.1% formic acid at a flow rate of 0.8 mL/min: 5–100% B (20 min). The sample concentration was 1 mg/mL, and the injection volume was 2 µL. Analytes were detected with a Dionex Ultimate DAD-3000 RS (wavelength range of 200–400 nm) and a Bruker Daltonics micrOTOF-QII quadrupole/time-of-flight (QTOF) mass spectrometer (Bremen, Germany) equipped with an Apollo electrospray ionization source in positive mode at 5 Hz over a mass range of *m/z* 50–1000 using the following instrument settings: nebulizer gas nitrogen, 5 bar; dry gas nitrogen, 9 L/min, 220 °C; capillary voltage 4500 V; end plate offset −500 V; transfer time 70 µs; collision gas nitrogen. Collision energy and collision RF settings were combined for each single spectrum of 1000 summations as follows: 250 summations with 20% base collision energy, 130 Vpp + 250 summations with 100% base collision energy, 500 Vpp + 250 summations with 20% base collision energy, and 130 Vpp + 250 summations with 100% base collision energy and 500 Vpp. Base collision energy was 50 eV for precursor ions with a *m/z* less than 500 and then linearly interpolated against *m/z* up to a maximum of 70 eV for precursor ions with a *m/z* of up to 1000. Internal dataset calibration (HPCmode) was performed for each analysis using the mass spectrum of a 10 mM solution of sodium formate in 50% isopropanol that was infused during LC re-equilibration using a divert valve equipped with a 20 µL sample loop. Bruker Compass Data Analysis software (v. 4.0, Bruker Daltonics) was used to analyze the LC-MS results.

### 3.2. Plant Collection and Authentication

Stem bark of *F. sagittifolia* was collected in Ikire, Osun State, Nigeria, in April 2018. The plant was identified and authenticated by Dr. S.A. Odewo at the Forestry Research Institute of Nigeria (FRIN), Ibadan. A voucher specimen was deposited at the FRIN herbarium, Ibadan, as FHI 111988 for reference use.

### 3.3. Extraction and Isolation

Air-dried and pulverized *F. sagittifolia* stem bark (1 kg) was macerated in ethanol for 72 h. The ethanol extract (51 g), obtained after concentrating under reduced pressure using rotary evaporator, was further subjected to liquid–liquid partitioning to obtain the hexane (8 g), ethyl acetate (11 g), and ethanol (31 g) fractions. The ethyl acetate fraction was subjected to vacuum liquid chromatography on 360 g silica gel with a gradient elution of n-hexane/dichloromethane (100:0 → 0: 100), dichloromethane/ethylacetate (100:0 → 0: 100), and ethylacetate/ethanol (100:0 → 40: 60); 200 mL of the eluents was collected for each solvent system, which afforded fourteen fractions, which were pooled together, according to their similar TLC profiles, to give eight fractions: F1 (1.1 g), F2 (0.6 g), F3 (0.4 g), F4 (2.7 g), F5 (1.7 g), F6 (1.0 g), F7 (0.8 g), and F8 (1.4 g). Further, 1.1 g of F5 was subjected to preparative HPLC using a binary mixture of acetonitrile and water (0.1% TFA) by gradient elution (30:70 → 100:0 in 40 min) at a flow rate of 10 mL/min to afford a mixture of compounds **1** and **2** (t_R_ = 14.0 min, 3.6 mg), **3** (t_R_ = 16.0 min, 2.1 mg), and mixture of compounds **4** and **5** (t_R_ = 17.9 min, 1.5 mg). Preparative HPLC of 0.9 g of F6 using a binary mixture of methanol and water (0.1% TFA) by gradient elution (20:80 → 100:0 in 60 min) at a flow rate of 10 mL/min afforded compound **6** (t_R_ = 35.4 min, 2.0 mg). Likewise, the part of F7 soluble in acetonitrile/water 50:50 (0.3 g) was subjected to preparative HPLC using a binary mixture of acetonitrile and water (0.1% TFA) by gradient elution (5:95 → 100:0 in 30 min) at a flow rate of 10 mL/min that afforded compound **7** (t_R_ = 15.0 min, 8.3 mg).

### 3.4. Compounds Identification

All compounds are known and have been identified by comparison with literature data. Identification of compounds **1**–**7** was based on comparison with published NMR data.

### 3.5. Spectral Data of Compounds **1**–**7**

**(2*R*)-(-)-Eriodictyol (1)** HR-ESI-MS *m/z* 289.0713 [M + H]^+^; LC-UV(MeOH): 220 nm and 288 nm; CD (c = 0.069 mM, MeOH) λ nm (∆ε) 260.5 (1.40), 229 (−0.73), 200 (−0.57, last reading); note that the data were obtained with a mixture of **1** with the achiral compound **2**. Therefore, the intensity of the CD bands is lower than it would be in case of pure **1**. The CD data are in full agreement with the presence of the *R*-enantiomer [16] (see also main text). ^1^H-NMR (600 MHz, (CD_3_OD) δ (ppm): 6.94 (1H, dd, *J* = 1.52 Hz, 0.85, H-5′), 6.81 (2H, d, *J* = 0.75 Hz, H-2′, H-6′), 5.93 (1H, d, *J* = 2.12 Hz, H-8), 5.91 (1H, d, *J* = 2.18 Hz, H-6), 5.31 (1H, dd, *J* = 12.75 Hz, 3.05 Hz, H-2), 3.10 (1H, dd, *J* = 17.06 Hz, 12.77 Hz, H-3ax), 2.73 (1H, dd, *J* = 17.12 Hz, 3.08 Hz, H-3eq); ^13^C-NMR (150 MHz, (CD3OD) δ (ppm): 197.4 (C-4), 167.9 (C-7), 165.6 (C-5), 165.0 (C-8a), 146.5 (C-4′), 115.8 (C-5′), 131.4 (C-1′), 114.3 (C-2′), 146.1 (C-3′), 118.8 (C-6′), 102.9 (C-4a), 96.6 (C-6), 95.7 (C-8), 80.1 (C-2), 43.7 (C-3). The NMR data are in agreement with the literature [17].

**2‘-Hydroxygenistein** (**2)** HR-ESI-MS *m/z* 287.0555 [M + H]^+^; LC-UV(MeOH): 264 nm and 284 nm; ^1^H-NMR (600MHz, (CD_3_OD) δ (ppm): 8.04 (1H, s, H-2), 7.07 (1H, d, *J* = 8.28 Hz, H-6′), 6.42 (1H, d, *J* = 2.35 Hz, H-3′), 6.41 (1H, d, *J* = 2.47 Hz, H-8), 6.39 (1H, d, *J* = 2.16 Hz, H-5′), 6.26 (1H, d, *J* = 2.16 Hz, H-6); ^13^C-NMR (150 MHz, (CD_3_OD) δ (ppm): 182.3 (C-4), 165.6 (C-7), 163.3 (C-5), 161.3 (C-4′), 159.8 (C-2′), 159.3 (C-2), 157.4 (C-8a), 156.3 (C-3′), 132.8 (C-6′), 122.2 (C-3), 110.4 (C-1′), 107.7 (C-5′), 105.8 (C-4a), 103.9 (C-3′), 99.8 (C-6), 94.4 (C-8). The NMR data are in agreement with the literature [19].

**Erycibenin A (3)** HR-ESI-MS *m/z* 373.1308 [M + H]^+^; LC-UV(MeOH): 212 nm and 264 nm; ^1^H-NMR (600 MHz, (CD_3_OD) δ (ppm): 8.06 (1H, s, H-2), 7.37 (2H, d, *J* = 8.69 Hz, H-2′, H-6′), 6.84 (2H, d, *J* = 2.35 Hz, H-3′, H-5′), 6.43 (1H, s, H-8), 3.64 (1H, d, *J* = 10.04 Hz, 2.57 Hz, H-2′′), 3.07 (1H, d, *J* = 13.99 Hz, 2.55 Hz, H-1′′ax), 2.72 (1H, d, *J* = 2.61 Hz, 13.97 Hz, H-1′′eq), 1.26 (6H, s, *J* = 2.16 Hz, H-4′′, H-5′′); ^13^C-NMR (150 MHz, (CD_3_OD) δ (ppm): 182.4 (C-4), 164.5 (C-7), 161.1 (C-5), 158.8 (C-8a), 157.9 (C-4′), 154.7 (C-2), 131.4 C-2′, 6′), 124.7 (C-1′), 123.5 (C-3), 116.3 (C-3′, 5′), 111.5 (C-6), 106.2 (C-4a), 94.6 (C-8), 79.7 (C-2′′), 74.0 (C-3′′), 25.8 (C-1′′), 25.6 (C-4′′), 25.2 (C-5′′). The NMR data are in agreement with the literature [21].

**Genistein (4)** HR-ESI-MS *m/z* 271.0618 [M + H]^+^; LC-UV(MeOH): 260 nm and 320 nm; ^1^H-NMR (600 MHz, (CD_3_OD) δ (ppm): 8.07 (1H, s, H-2), 7.38 (2H, d, *J* = 8.63 Hz, H-2′, H-6′), 6.85 (2H, d, *J* = 8.67 Hz, H-3′, H-5′), 6.23 (1H, d, *J* = 2.08, H-6), 6.35 (1H, d, *J* = 2.20, H-8); ^13^C-NMR (150 MHz, (CD_3_OD) δ (ppm): 182.5 (C-4), 163.6 (C-7), 161.2 (C-5), 159.7 (C-8a), 159.5 (C-4′), 152.3 (C-2), 131.1 (C-2′), 124.9 (C-1′), 124.5 (C-3), 116.0 (C-3′), 104.6 (C-4a), 99.4 (C-6), 94.5 (C-8). The NMR data are in agreement with the literature [23].

**Moracin P (5)** HR-ESI-MS *m/z* 327.1248 [M + H]^+^; LC-UV(MeOH): 220 nm and 320 nm, ^1^H-NMR (600 MHz, (CD_3_OD) δ (ppm): 7.24 (1H, s, H-7), 6.89 (1H, d, *J* = 0.99 Hz, H-3), 6.86 (1H, m, H-4), 6.76 (2H, d, *J* = 2.18, H-2′, H-6′), 6.25 (1H, d, *J* = 2.18, H-4′), 3.80 (1H, dd, *J* = 7.60 Hz, 5.39 Hz, H-2′′), 3.13 (1H, m, H-1′′ax), 2.84 (1H, ddd, *J* = 16.18 Hz,7.63 Hz, 1.20 Hz, H-1′′eq), 1.37 (3H, s, H-5′′), 1.29 (3H, s, H-4′′); ^13^C-NMR (150 MHz, (CD_3_OD) δ (ppm): 159.7 (C-3′,C-5′), 156.3 (C-2), 152.3 (C-7a), 151.7 (C-6), 133.4 (C-1′), 121.5 (C-4), 117.4 (C-5), 124.5 (C-3a), 104.6 (C-2′, C-6′), 103.7 (C-4′), 103.4 (C-3), 99.4 (C-7), 78.0 (C-3′′), 70.3 (C-2′′), 32.2 (C-1′′), 25.7 (C-4′′), 20.8 (C-5′′). The NMR data are in agreement with the literature [24].

**Peucedanol (6)** HR-ESI-MS *m/z* 265.1109 [M + H]^+^; LC-UV(MeOH): 204 nm and 332 nm; ^1^H-NMR (600 MHz, (CD_3_OD) δ (ppm): 7.84 (1H, d, *J* = 9.50, H-4), 7.40 (1H, s, H-5), 6.73 (1H, s, H-8), 6.17 (1H, d, *J* = 9.49 Hz, H-3), 3.62 (1H, d, *J* = 8.43 Hz, H-2′), 3.08 (1H, d, *J* = 13.67 Hz, H-1′ax), 2.55 (1H, d, *J* = 14.01 Hz, 10.45 Hz, H-1′eq), 1.26 (3H, s, H-4′), 1.25 (3H, s, H-5′); ^13^C-NMR (150 MHz, (CD_3_OD) δ (ppm): 165.2 (C-2), 162.7 (C-7),157.0 (C-8a), 152.3 (C-3a), 145.8 (C-4), 131.2 (C-5), 127.0 (C-6), 114.4 (C-3), 111.8 (C-4a), 102.7 (C-8), 78.8 (C-2′), 73.4 (C-3′), 33.0 (C-1′), 25.2 (C-4′), 24.7 (C-5′). The NMR data are in agreement with the literature [26].

**Dihydrophaseic acid (7)** HR-ESI-MS *m/z* 281.1395 [M−H]^−^; LC-UV(MeOH): 268 nm; CD (c = 0.071 mM, MeOH) λ nm (∆ε) 266 (−3.8). The CD data are in full agreement with literature [29]. ^1^H-NMR (600 MHz, (CD_3_OD) δ (ppm): 7.99 (1H, *J* = 16 Hz, 0.8 Hz, H-7), 6.53 (1H, d, *J* = 16 Hz, H-8), 5.77 (1H, q, 1 Hz, H-10), 4.12 (1H, ddd (tt), *J* = 10 Hz, 7 Hz, H-3), 3.81 (1H, d, *J* = 7 Hz, 2 Hz, H-12β), 3.72 (1H, d, *J* = 7 Hz, H-12α), 2.09 (3H, d, *J* = 1.3 Hz, H-15), 2.04 (1H, ddd, *J* = 13.7 Hz, 7.0 Hz, 1.8 Hz, H-4α), 1.86 (1H, ddd, *J* = 13.6 Hz, 7.0 Hz, 1.8 Hz, H-2α), 1.74 (1H, dd, *J* = 13.7 Hz, 10.8 Hz, H-4β), 1.66 (1H, ddd, *J* = 13.6 Hz, 10.9 Hz, 2.2 Hz, H-2β), 1.15 (3H, s, H-14), 0.94 (3H, s, H-13).^13^C-NMR (150 MHz, (CD_3_OD) δ (ppm): 169.3 (C-11), 151.2 (C-9), 134.9 (C-7), 131.5 (C-8), 118.9 (C-10), 87.5 (C-5), 83.0 (C-6), 77.0 (C-12), 65.7 (C-3), 49.2 (C-1), 45.7 (C-4), 44.3 (C-2), 21.0 (C-15), 19.4 (C-14), 16.1 (C-13). The NMR data are in agreement with the literature [28] but need to be revised in some details (see main text).

### 3.6. Simulation of CD Spectra

Molecular models of (2*R*)-(−)-eriodictyol (**1**) and dihydrophaseic acid (**7**) were generated with the Molecular Operations Environment (MOE, v. 2020.09, Chemical Computing Group, Montreal, Canada) using the MMFF94x force field. A conformational search was conducted in each case using the systematic search method and the same force field. The lowest-energy conformers within an energy window of 3 kcal/mol above the global minimum were energy minimized with the semi-empirical method AM1. In case of **1**, the two lowest-energy conformers were used for the spectra simulation. In case of dihydrophaseic acid **7**, only one model for the lowest energy conformer was analyzed, as described.

The structures were saved in mol2-format and imported into Gaussian (v. Gaussian03W, Gaussian Inc., Pittsburgh, PA, USA), where they were fully energy minimized using density functional theory (DFT) with the B3LYP density functional and the 6-31G(d,p) basis set. Time-dependent DFT (TDDFT) calculations using the same functional and basis set were then performed on the minimized structures. In all cases, the first (i.e., lowest in frequency) 30 electronic transitions were computed. The output from Gaussian for electronic transition energies (E in eV) and rotator strengths (R, dipole length in cgs) was extracted, and ECD spectra were simulated by applying a Gaussian shape function with a band width at 1/e height σ = 0.15 and 0.20 eV, respectively, for compounds **1** and **7**. In case of (*S*)-**1**, the simulated spectra of two conformers with very similar energy (ΔE = 0.33 kcal/mol) were Boltzmann-averaged based on the computed total energy of each conformer in a ratio of 0.64:0.36. The simulated spectra of (2*S*)- and (2*R*)-eriodictyol shown in Figure 2 together with the experimental spectrum of **1** (in the 30:70 mixture with CD-inactive **2**) were scaled by factor 0.15. In case of **7,** the spectrum was simulated only for one single lowest-energy conformer in the manner described, but no intensity scaling was necessary to obtain the plot shown along with the experimental CD curve in Figure 3. No shift on the energy scale had to be applied in either case.

### 3.7. Antimicrobial Activity Bioassay

Minimum inhibitory concentration (MIC) and minimum bactericidal/fungicidal concentration (MBC/MFC) of compounds **1**–**7** were determined using the broth micro-dilution method (96-well plates).

#### 3.7.1. Microbial Cultures

Five bacteria (*Staphylococcus aureus* (ATCC 6571), *Escherichia coli* (ATCC 700728), *Pseudomonas aeruginosa* (ATCC 27853), *Salmonella typhi* (ATCC 14028), and *Klebsiella pneumoniae* (ATCC 700303) and two fungi: *Candida albicans* (Clinical isolates) and *Aspergillus niger* (Clinical isolates)) were provided by the Pharmaceutical Microbiology Department, University of Ibadan, Ibadan, Nigeria. All the microbes were sub-cultured from the original culture and incubated overnight at 37 °C for 24 h and at 25 °C for 48 h in the case of bacteria and fungi, respectively. The standard drugs, gentamycin and ketoconazole, were used as the positive controls for the bacteria and fungi, respectively, while 1% DMSO was used as a negative control.

#### 3.7.2. Minimum Inhibitory Concentration (MIC) and Minimum Bactericidal/Fungicidal Concentration (MBC/MFC)

Each compound was dissolved separately in 1% DMSO to obtain a stock solution of 1 mg/mL. This was serially diluted via two-fold dilution to 0.5–0.0039 mg/mL using tryptic soy broth. Each of the microplate wells was inoculated with 10 µL of the test organisms’ cultures and incubated at 37 °C and 25 °C for 24 h and 48 h for bacteria and fungi, respectively. The lowest concentration that showed no growth or turbidity after the required incubation was taken as the MIC.

After incubation, 10 µL of 0.2 mg/mL *p*-iodonitrotetrazolium was added to each well and incubated for another 30 min. Wells with a color change from yellow to pinkish red were indicative of microbial growth. The lowest concentration that showed no trace of color change was taken as the MBC/MFC. Gentamycin and ketoconazole were used as the positive controls (standard drugs) for bacteria and fungi, respectively. Both standards were also serially diluted to their respective concentrations and tested in the same manner as described above.

## 4. Conclusions

Seven known compounds ((2*R*)-eriodictyol, 2′-hydroxygenistein, erycibenin A, genistein, moracin P, peucedanol, and dihydrophaseic acid) were isolated from the stem bark of the title plant and are reported for the first time as constituents of *F. sagittifolia*. Moracin P, known from other Moraceae, is reported for the first time from the genus *Ficus*. These isolated compounds justified the taxonomic placement of *F. sagittifolia* in the genus *Ficus* and, therefore, can be used as a chemotaxonomic marker for *F. sagittifolia*. The isolated compounds showed moderate activity against the tested bacteria and fungi compared to the reference drugs (gentamycin and ketoconazole). The isolated mixture of genistein and moracin P displayed strong activity against *K. pneumoniae,* while dihydrophaseic acid was the most active against *P. aeruginosa* and *A. niger*.

These compounds might possess a potential as starting points for the development of antimicrobial drugs, and *F. sagittifolia* could, hence, be considered as a potential source of such compounds with interesting antimicrobial activity.

## Figures and Tables

**Figure 1 plants-12-02801-f001:**
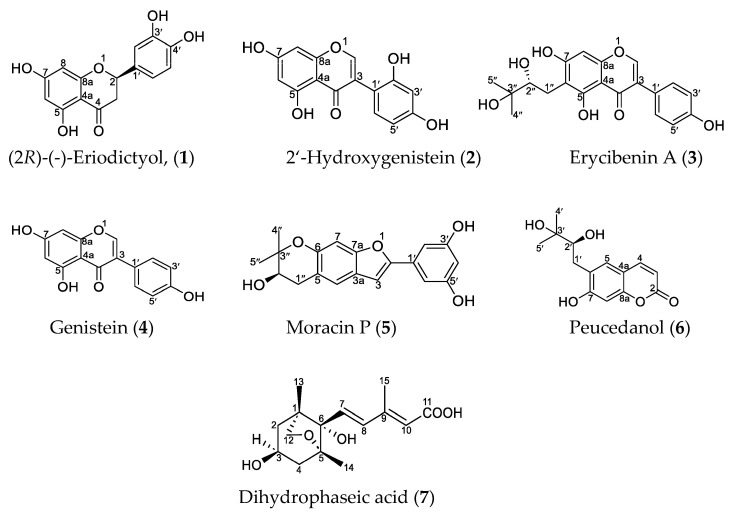
Compounds **1**–**7** isolated from *F. sagittifolia* stem bark.

**Figure 2 plants-12-02801-f002:**
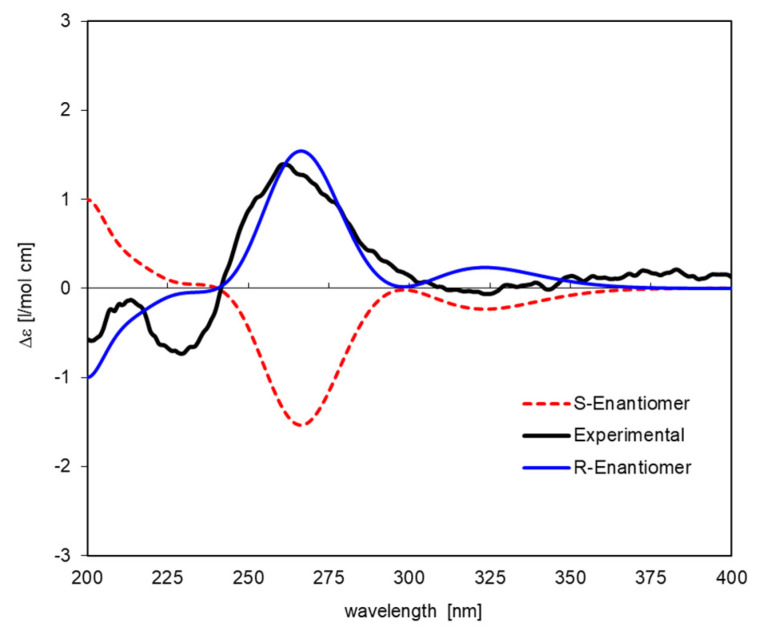
Experimental (black curve) and simulated (TDDFT at the B3LYP/6-31G(d,p) level of theory) CD spectra of the (2*S*)-(+)-enantiomer (red dotted curve) and of the (2*R*)-(−)-enantiomer (blue curve) of Eriodictyol (**1**).

**Figure 3 plants-12-02801-f003:**
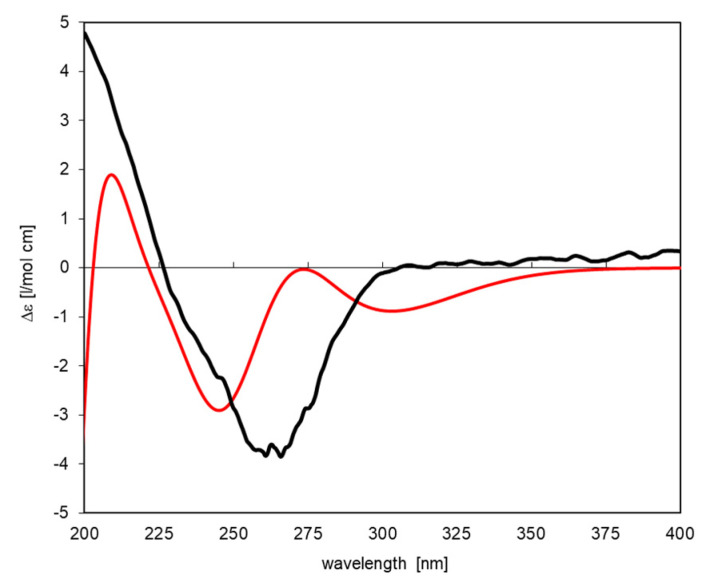
Experimental (black curve) and simulated (TDDFT at the B3LYP/6-31G(d,p) level of theory) CD spectra of dihydrophaseic acid (**7**) (red curve).

**Table 1 plants-12-02801-t001:** MIC and MBC/MFC of compounds **1–7** from *F. sagittifolia* stem bark (all data are expressed in mg/mL).

	Compounds													
Microorganism	1 + 2 (30:70)		3		4 + 5 (80:20)		6		7		Gentamycin		Ketoconazole	
	MIC	MBC/ MFC	MIC	MBC/ MFC	MIC	MBC/ MFC	MIC	MBC/ MFC	MIC	MBC/ MFC	MIC	MBC/ MFC	MIC	MBC/ MFC
*S. aureus*	0.25	0.25	0.25	0.25	0.25	0.25	0.25	0.25	0.25	0.25	0.005	0.005	NA	NA
*E. coli*	0.0625	0.25	0.0313	0.25	0.25	0.25	0.25	0.25	0.125	0.125	0.005	0.01	NA	NA
*P. aeruginosa*	0.125	0.25	0.0625	0.25	0.156	0.25	0.125	0.25	0.0078	0.25	0.01	0.01	NA	NA
*S. typhi*	0.125	0.25	0.125	0.25	0.25	0.125	0.25	0.25	0.25	0.25	>0.01	>0.01	NA	NA
*K. pneumoniae*	0.125	0.125	0.125	0.125	<0.0039	0.0313	0.25	0.125	0.125	0.25	0.01	0.01	NA	NA
*C. albicans*	0.125	0.25	0.125	0.25	0.125	0.5	0.0625	0.5	0.125	0.125	NA	NA	0.01	0.01
*A. niger*	0.0625	0.125	0.0313	0.25	0.0156	0.125	0.0313	0.125	<0.0039	0.25	NA	NA	0.005	0.005

MIC = Minimum Inhibitory Concentration; MBC/MFC = Minimum Bactericidal Concentration/Minuimum Fungicidal Concentration; *S. auerus* = *Staphylococcus aureus*; *E. coli* = *Escherichia coli*; *P. aeruginosa* = *Pseudomonas aeruginosa*; *S. typhi* = *Salmonella typhi*; *K. pneumoniae* = *Klebsiella pneumoniae*; *C. albicans* = *Candida albicans*; *A. niger* = *Aspergillus niger*; NA = Not applicable.

## Data Availability

Not applicable.

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
