# Peer review of "Chemical Constituents from Ficus sagittifolia Stem Bark and Their Antimicrobial Activities"

_plants, 2023, doi:10.3390/plants12152801_

Round 1

Reviewer 1 Report

Interest in plants as sources of antimicrobial agents continues unabated. Part of the population of developing countries has used plants since ancient times to treat infectious and other diseases. In this regard, this work is of interest. Although isolated compounds are known and there is no great novelty, the work reveals several new facts. Moracin in the genus Ficus is listed for the first time. Also, for the first time, seven known compounds have been reported as constituents of Ficus sagittifolia. Determination of minimal bactericidal or fungicidal inhibitory concentrations of isolated compounds is valuable and this article deserves publication. 

The article is well written. I did not find any serious blots in the language.

Author Response

Interest in plants as sources of antimicrobial agents continues unabated. Part of the population of developing countries has used plants since ancient times to treat infectious and other diseases. In this regard, this work is of interest. Although isolated compounds are known and there is no great novelty, the work reveals several new facts. Moracin in the genus Ficus is listed for the first time. Also, for the first time, seven known compounds have been reported as constituents of Ficus sagittifolia. Determination of minimal bactericidal or fungicidal inhibitory concentrations of isolated compounds is valuable and this article deserves publication. 

Reply:

We thank the reviewer for her/his time to evaluate our manuscript and the positive feedback.

Reviewer 2 Report

The manuscript described the isolation of 7 well-known compounds from Ficus sagittifolia stem bark along with their antimicrobial activities. The manuscript include already reported compounds as well as the compounds not totally isolated from each other but there are compounds as mixtures. I think the authors need to modify this work by increasing of new data such as 

1. Chemotaxonomy study

2. Chemometric analysis (PCA and AHC) of the data for supporting of the chemotaxonomy

3. The docking of the compounds for suggesting of the mechanism of the compounds actions

4. The possible biosynthetic pathway of the isolated compounds

5. The theoretical NMR study of these compounds

6. The experimental and DFT calculated studies of isolated compounds

7. Structure activity relationship (SAR) study

OK

Author Response

The manuscript described the isolation of 7 well-known compounds from Ficus sagittifolia stem bark along with their antimicrobial activities. The manuscript include already reported compounds as well as the compounds not totally isolated from each other but there are compounds as mixtures. I think the authors need to modify this work by increasing of new data such as 

  1. Chemotaxonomy study
  2. Chemometric analysis (PCA and AHC) of the data for supporting of the chemotaxonomy
  3. The docking of the compounds for suggesting of the mechanism of the compounds actions
  4. The possible biosynthetic pathway of the isolated compounds
  5. The theoretical NMR study of these compounds
  6. The experimental and DFT calculated studies of isolated compounds
  7. Structure activity relationship (SAR) study

Reply:

The reviewer asks for various further investigations which are impossible to provide at this stage:

  1. Some chemotaxonomic implications, as far as can be made safely on grounds of the data, are already mentioned in the manuscript (pp 4, 5; lines144-158). Further discussion on chemtaxonomy would inevitably lead to more speculative conclusions which we do not find useful.
  2. It is not possible, not within the scope of the present work and not useful on grounds of the present data to apply chemometric methods (PCA etc.). This would require metabolic profiling of many samples from the title plant (e.g. from various locations, different plant parts etc.) as well as other species. It might, e.g., be interesting to compare many Ficus species in such an approach but this is, clearly, not the scope of the present work, which was primarily to find antimicrobial compounds in the bark of the title species.

3-7. In the same way, the reviewer asks for various items which would make completely different studies which were not within our scope.

  1. Docking studies are only useful if a biological target is known and its 3D structure has been determined. In the present case, neither is the case so such studies will have to await results from structural biology-oriented work which is beyond the scope of our phytochemical study.
  2. The general biosynthesis of flavonoids and some other metabolites found in our study is known so merely discussing it would not add significantly to our study. As already stated, this work aimed at finding compounds with antimicrobial activity in the title plant and was not about biosynthesis of these compounds.
  3. The NMR spectra were used successfully to identify the structures of the isolated compounds. Any theoretical studies on NMR are unnecessary in such a case.
  4. Computational studies using DFT were conducted in two cases where it appeared necessary and useful to fully clarify the structures. It does not appear useful to apply such methods where the structure is in complete agreement with unambiguous previous reports.
  5. Structure-activity relationship studies are not useful on grounds of a data set such as the one presented here. It consists of just a small number of compounds of different chemical classes; it is not useful to compare structures and activities in such a case because (a) the structures are not similar enough and (b) it is not proven (and in fact rather unlikely) that the compounds act by the same mechanism. SAR-studies may become useful once more structurally related compounds are tested in the future. However, this is beyond the scope of the present report.

Thus, we can only state that the reviewer has recognized various possibly interesting starting points for further, following investigations. However, we do not see the necessity to add or otherwise modify the present manuscript on these grounds.

We thank the reviewer for her/his time to evaluate our manuscript.

Reviewer 3 Report

The authors reported the the isolation of 7 compounds from the ethylacetate fraction of an ethanolic extract of the stem bark of Ficus sagittifolia and their antimicrobial activity against various pathogens. The experimental design is adequate, the results are clearly presented and the conclusions are consistent with them. Moreover, the authors gave additional information about the correct assignment of some protons in the structure of compound 7 and some taxonomic conclusions. I consider that this manuscript can be accepted for publication in this journal after a minor revision.

Remarks:

1. Fig. 1. Resize the numbers of Carbon atoms and put them inside the structure, where it is possible;

2. The name of compound 2 should start with capital letter (Fig. 1).

3. Ln. 255-263: Did you use the whole amount of fraction F5, F6 and F7 for isolation of the compounds or a portion of them? Please, clarify and specify.

Author Response

The authors reported the the isolation of 7 compounds from the ethylacetate fraction of an ethanolic extract of the stem bark of Ficus sagittifolia and their antimicrobial activity against various pathogens. The experimental design is adequate, the results are clearly presented and the conclusions are consistent with them. Moreover, the authors gave additional information about the correct assignment of some protons in the structure of compound 7 and some taxonomic conclusions. I consider that this manuscript can be accepted for publication in this journal after a minor revision.

Remarks:

  1. Fig. 1. Resize the numbers of Carbon atoms and put them inside the structure, where it is possible;

Reply: All atom labels reduced in size and moved closer to their appropriate positions.

  1. The name of compound 2 should start with capital letter (Fig. 1).

Reply: This was corrected.

  1. Ln. 255-263: Did you use the whole amount of fraction F5, F6 and F7 for isolation of the compounds or a portion of them? Please, clarify and specify.

Reply:

The whole amount of F5 was 1.7 g, for the purification using HPLC, 1.1 g was used as now stated in the manuscript.

The whole amount of F6 was 1.0 g, for the purification using HPLC, 0.9 g was used as now stated in the manuscript.

The whole amount of F7 was 0.8 g, but only 0.3 g dissolved in the solvent (Acetonitrile and water 50:50) before the HPLC analysis and so 0.3 g was used for the purification using HPLC as now stated in the manuscript.

We thank the reviewer for her/his time to evaluate our manuscript and the positive feedback.

Round 2

Reviewer 2 Report

The authors did not provide any acceptable responses. The authors should know

1. The PCA can be used with you data as comparison with the collective data of previous works

2. The docking studies can be performed for the active compounds against the bacteria they found by selecting the proteins of these microbes

3. The SAR can be easily done on your work where you have already active and non active compounds 

4. The theoretical NMR and DFT studied not for elucidation of you compounds, you compounds are already well known, but for studying of the chemo-biological relationship 

finally, I think this manuscript is still very week to be suitable for publication in plants. Plants is a Q1 journal!!!!!

Ok

Author Response

Thank you again for your time and effort.

The answer in this case is addressed directly to the editor.